# Farmers’ Attitudes in Connection with the Potential for Rodent Prevention in Livestock Farming in a Municipality in North Rhine-Westphalia, Germany

**DOI:** 10.3390/ani13243809

**Published:** 2023-12-10

**Authors:** Anna Schulze Walgern, Odile Hecker, Bernd Walther, Marc Boelhauve, Marcus Mergenthaler

**Affiliations:** 1Department of Agriculture, South Westphalia University of Applied Sciences, 59494 Soest, Germany; 2Institute for Plant Protection in Horticulture and Forests, Vertebrate Research, Julius Kühn-Institute, Toppheideweg 88, 48161 Münster, Germany

**Keywords:** rodent-prevention potential, one health, livestock farming, farmers’ attitude, theory of planned behaviour

## Abstract

**Simple Summary:**

Rodents threaten the one health approach in livestock farming. In the present observational case study, livestock farmers collaborated with a pest controller in a pilot project in North Rhine-Westphalia, Germany. The objective was to evaluate factors influencing rodent-prevention potential on 24 livestock farms after 1.5 years project duration. Farmers’ attitudes were surveyed at the project’s start in March 2019. Potential for rodent prevention was assessed by an external expert in 2020. About half of the farms showed good potential for rodent prevention. Willingness to change at the project start played a central role for rodent-prevention potential. The study underscores the necessity of better informing farmers about rodent control and prevention, emphasizing the importance of preventive measures for long-term rodent control. It also highlights the need for improved advisory services for farmers.

**Abstract:**

Rodents in livestock farming constitute a threat to the one health approach. In the present observational case study, livestock farmers worked together with a pest controller within a pilot project. The aim of the study was to assess determinants associated with rodent-prevention potential. The study started in March 2019 on 24 livestock farms in a municipality in North Rhine-Westphalia/Germany. At the beginning of the project a survey on the determinants expected to be related to prevention potential was conducted. To determine the potential for rodent prevention, an expert person, who was not involved in the project before, assessed the on-site conditions of the farms after 1.5 years of project duration in 2020. The potential for rodent prevention was good for about half of the farms. There were significant differences in the willingness to make changes at the project’s start between farms with a high and a low potential for rodent prevention after 1.5 years. There is a general need for action to provide farmers with more practical information on rodent control. This is aggravated by the insufficient advisory services offered to farmers. This study confirms the importance of implementing preventive measures in the control of rodents to ensure that anticoagulant rodenticides are handled responsibly to reduce the impact on non-target species.

## 1. Introduction

Due to good living conditions related to the availability of feed, water, and shelter, rodents are widespread on livestock farms. Rodents cause direct feeding loss, contamination of feeding stocks by urine and faeces, and damage to infrastructures [1,2,3]. Within a one health approach rats, especially Norwegian rats (*Rattus norvegicus*), are also vectors for the transmission of various zoonotic infectious agents and parasites to other livestock (e.g., *leptospirosis*), pets (e.g., *toxoplasmosis*), and humans (e.g., *hantavirus*) [4,5]. The presence of rodents can cause stress and discomfort among livestock. Rodents can startle animals, disrupt their feeding and resting behaviours, and create an unsanitary environment, affecting the overall well-being and productivity of livestock. The damage caused by rodents in terms of reduced feed quality, structural damage, increased veterinary costs due to disease transmission, and productivity losses can reduce the profitability of intensive livestock farming operations [2,3,6]. To protect livestock, rodent control is obligatory on livestock farms (e.g., in Germany the hygiene regulations for pig farming “Schweinehaltungshygieneverordnung” [7]). Farmers usually carry out prevention and control measures on their own, including the use of rodenticides [8]. However, rodent-control measures on livestock farms are often not implemented regularly [8,9].

To avoid environmental risks in the use of anticoagulants, risk minimization measures must be implemented. This includes restricting use of rodenticides to professionals with proof of expertise [10]. Products are divided into first-generation anticoagulant rodenticides (FGARs) with active substances, such as warfarin, coumatetralyl, chlorphacinone, and higher potent second-generation anticoagulant rodenticides (SGARs) with active substances such as bromadiolone, difenacoum, brodifacoum, flocoumafen, and difethialone. Nevertheless, resistance to rodenticides with anticoagulant substances has already been detected in many European regions [11]. All anticoagulants currently approved as biocidal active ingredients in the EU are classified as toxic to reproduction and specific target organ toxicity [12,13]. Therefore, they are dangerous to the environment and can also endanger non-target species such as foxes, otters, owls, and other birds [14,15,16,17,18,19]. Due to the highly toxic effect and persistence of SGARs, the implementation of preventive measures should be emphasized on livestock farms.

Numerous previous studies in different countries have shown that farmers have negative attitudes towards rodents in farms and that they take different actions to keep rodents away from their livestock [20,21,22,23,24,25,26]. Cooperation with a professional pest controller can have a positive impact on perceptions and attitudes towards the implementation of prevention and control measures [27]. In order to increase the willingness to implement rodent-control measures, it is important to identify attitudinal determinants related to rodent prevention and control. This understanding can lead to improving impediments and lifting barriers to uptake.

In this paper, we build on the theory of planned behaviour (TpB) to identify attitudinal determinants on rodent-prevention potential on livestock farms. The TpB emphasizes individuals’ intentions, which are influenced by attitudes toward the behaviour (e.g., using rodent-control measures), subjective norms (perceptions about what others consider appropriate), and perceived control over the behaviour (perceived barriers and facilitators). The investigation was implemented within a case study of a specific project to improve rodent control with the help of rodent-control professionals. To measure the potential for rodent prevention, an expert who was not involved in the project but who had the necessary expertise was used to independently assess the condition of the farms to prevent rodent infestation after 1.5 years project duration. We hypothesized that different attitudes of farmers measured at the onset of the project would determine the level of rodent-prevention potential on livestock farms after 1.5 years. A better understanding of attitudinal determinants will potentially help to improve rodent control in livestock farming. This will potentially contribute to reduce zoonotic agents, improve the health status and the performance of livestock, and, thereby, contribute to a one health approach.

## 2. Theoretical Background

The importance of socio-psychological factors for farmers’ adoption of new agricultural practices is highlighted comprehensively in a meta-analysis [28]. The theory of planned behaviour (TpB) is a model for predicting behaviour and is used to explain the psychological factors that influence it [29]. In the present study, the TpB was extended to identify attitudinal determinants related to the potential for rodent prevention on livestock farms. The TpB has been used in social science research in agriculture, especially in biosecurity on livestock farms, e.g., [30]. While other behavioural models have also been employed to explain farmers’ biosecurity measures, e.g., [31], the main advantage of the TpB over other behavioural theories lies in its explicit focus on intention as a direct precursor to behaviour. Compared to other behavioural models, the application of the TpB in rodent management differs as it specifically emphasizes the role of attitudes, social influences in the form of subjective norms, and perceived behavioural control in predicting and influencing behaviours related to rodent-control practices [32]. By understanding and addressing these factors, TpB could offer insights into tailoring interventions and strategies that effectively promote the adoption of appropriate rodent-control measures based on individuals’ beliefs, social influences, and perceived control, potentially leading to more effective and sustained behaviour change in this context.

According to Ajzen [29], attitude is a person’s own evaluation of the behaviour and can be negative or positive. This means that the more positive the attitude, the stronger the intention is to carry out the behaviour. Studies have also found a correlation between attitudes towards certain biosecurity management practices on livestock farms and the intention to implement these measures in the livestock sector [33,34,35,36].

The second influencing factor in the TpB is the subjective norm. It reflects the perceived social pressure from one’s own social environment [32]. This means that people rather tend to perform actions when they think that other people evaluate them positively. This is probably due to the fact that it motivates individuals to orientate themselves towards significant others [32]. Studies showed that subjective norms are also related to the implementation of certain biosecurity measures [34,35,36,37,38].

The implementation of a certain behaviour is also influenced by internal and external factors. Here, assessments of the controllability of one’s own behaviour also play an important role [32]. Perceived behaviour control is the assessment of how easy or difficult it is to carry out a certain behaviour [39]. In the context of biosecurity in livestock farming, studies have shown that perceived behaviour control can be an influencing factor [35,36,40]. However, the perceived behaviour control of certain biosecurity measures depends on the perceived severity of the potential disease [40], the information available, and the costs involved in implementation [36].

Further studies [41], have examined complementary factors that may be related to the implementation of biosecurity measures. In the present study, the model of the TpB was extended to include these additional factors as follows: Sok [42] investigated the decision to participate in a voluntary vaccination programme. The study shows that risk perception is an important influencing factor. Other studies also found that problem or risk awareness influence the willingness for implementation [40,43,44,45]. According to Mankad [46], due to a close link between emotions and attitudes, emotions can also play a central role in decision making on the implementation of biosecurity measures. On the other hand, a lack of knowledge can be a reason for a low willingness for implementation [35,47]. However, Nöremark et al. [48] point out that knowledge does not necessarily lead to changes in behaviour. Further studies have shown that entrepreneurial behaviour is also influenced by the tendency for innovation [49,50].

## 3. Data and Methods

In the present observational study, livestock farmers in a selected municipality in Germany worked together with a pest controller in order to increase rodent-prevention potential. It was an areal approach with an administratively, clearly defined geographic delineation in order to take advantage of farm–neighbourhood effects in rodent control. Within the framework of the project, monitoring and control of an acute rodent infestation was contracted and delegated by hiring a professional pest controller. However, the responsibility to organise the farm with preventive measures in such a way that a settlement of rodents can be reduced in the long-term remained with the farmer. The pest control and prevention measures in the project were accompanied by several scientific investigations. The professional pest controllers hired by the farmers were partially financed by the animal disease fund of the German federal state North Rhine-Westphalia (TSK NRW) with decreasing financial support over the course of the project [51].

The project was restricted to one municipality in North Rhine-Westphalia/Germany. Therefore, our study has an observational case study character whereby the implementation was continuously monitored. The research design did not allow for the comparison of a somehow defined treatment with a control or a clearly defined before-situation with an after-situation. It was a multi-farm approach, so that prevention and control measures could be coordinated in the neighbourhood [52]. During the project, surveys and farm visits took place in the first one and a half years. The data from three different surveys at the different points in time were included in the present analyses.

### 3.1. Selection of the Project Municipality and Project Farms

The project municipality was multi-stage selected based on certain characteristics to ensure the presence of different farm types: there should be at least 80 and at most 135 livestock farms, at least 25 pig farms, at least 25 cattle farms, and at least one poultry farm in the municipality. Furthermore, the number of human inhabitants should not exceed 30,000 in order to ensure that the project is implemented in a rural area. We identified 22 of the 396 municipalities (including cities) in North Rhine-Westphalia as meeting these criteria. Out of these 22 municipalities, 5 were selected randomly. In these five municipalities, a mail-based written initial survey was used to assess the status of rodent control. The municipality with the highest number of returned mail-based questionnaires, i.e., a response rate of 26%, was selected as the project municipality. The selected municipality was located in the district of Steinfurt at the northern border of the state of North Rhine-Westphalia bordering the federal state of Lower Saxony.

As part of the project, farmers were informed about the project via two kick-off events in October and November 2018. Farmers who wished to participate in the project signed a declaration of commitment, in which the conditions for participation were regulated and by which farmers gave their informed consent to the project. In order to ensure that the sampled project farms did not include too many small-structured livestock farms, farms that fulfilled certain minimum production units were taken into account in the selection procedure of the present study (Table 1).

In the selected municipality, 24 of 102 [53] farmers decided to take part in the project [51]. These 24 project farms took advantage of the option to commission a listed pest controller who cooperated with the project farmer. For the 24 farms, all animals were at least partially housed in sheds. Pigs were completely indoors. For cattle, some farms had seasonal free grazing in pasture. All animals were fed in troughs. Cooperating pest controllers had to meet predefined quality criteria such as being licenced for their activities and having some minimum control intervals.

The pest controllers’ working methods differed regarding inspection intervals ranging from 7 to 18 inspections in the first 1.5 years. Some pest controllers made a distinction between acute and routine control visits (e.g., weekly over a short period or longer intervals), while others always had a fixed interval (e.g., every two months). The working method was similar with regard to the use of different active ingredients and bait materials. The working methods of the pest controllers tended to vary from farm to farm, which indicates an individualised control strategy. All pest controllers used common active ingredients and also changed them; brodifacoum was only used to a lesser extent and only for a limited period for acute control. Bait boxes were placed at similar locations in the livestock farms indicating that the respective hotspots were assessed similarly.

### 3.2. Collection of Data

In order to identify the attitudinal determinants associated with the potential for rodent prevention, questionnaire-based face-to-face interviews were conducted on 24 farms at the beginning of the project period. The questionnaire was developed according to the TpB (Table A1 in Appendix A). Further determinants were added from literature research, which were bundled in the underlying model (Figure 1). The focus of the survey was to identify attitudinal determinants that may be related to the implementation of control and prevention measures. Interviews with project participants took place on-site between March and July 2019 and lasted on average 44 min (±18 min). The survey was conducted as a tablet-based interview with closed and open questions. It focused on end-point scales from 1 to 10 and Likert scales from 1 to 5 or ranking questions. Thus, the closed questions could be answered directly in the questionnaire on the tablet. Complementary statements made by the farmers were written down. All information was collected through interviews at this stage of the project, i.e., rodent infestation and prevention potential were not directly observed on-site of the farms.

After one year of project duration, the project participants were surveyed again. In a paper–pencil survey, farmers were asked to assess their satisfaction with the project and the likelihood of working with their pest controller after the end of the project. The survey took place in January 2020. For this purpose, all project participants were contacted by mail and asked to complete a two-page questionnaire which contained open and closed questions. The focus of the present study is on the closed questions, which were answered on a Likert scale from 1 to 5. Only the following questions were used in the present research: “*How likely are you to continue working with your pest controller after the end of the project?*” and “*Overall, how satisfied are you with the success of the pest control measures?*”.

After 1.5 years of project duration, after a stable trust basis for on-site visits had been established, all participating farms were visited from June to August 2020. During these farm visits a project-independent expert person assessed the on-site rodent infestation and the habitat quality of farm structures to estimate the risk of rodent infestation and the success of the control measures introduced by the pest controller based on an endpoint-based questionnaire. Farm visits lasted approximately one hour. Aspects important to the expert knowledge of the independent person were considered holistically and led to an overall score (scale 1 to 5). Although multiple aspects were considered, they were not scored separately. Focus of the present study is the statement on the potential for rodent prevention “*The farm is in such a condition that prevents rodents from settling to the maximum possible extent*”. This statement was given by the project-independent expert person based on his farm visit, i.e., independent of the farmers’ own assessment. Whereas most studies employing the TpB rely on behavioural intention as the dependent variable, we used the outcome of actual behaviour at the farm level [28].

### 3.3. Data Analysis

Data were first analysed using simple descriptive statistics (IBM SPSS Statistics 25). In the evaluation of endpoint scales, medians, means, and standard deviations were calculated assuming equidistance of the scales. Frequencies were determined for categorical variables. The ranking questions were converted so that a rank was available for each answer category. If a category was not selected by a participant, this was equivalent to a rank of 0. Subsequently, the mean value and standard deviation were determined assuming equidistance between the individual ranks. Open questions were categorised by content analysis and evaluated quantitatively so that frequencies could be calculated.

Individual items were summarised by taking the mean of several items from a particular factor. Furthermore, the factors in connection with the potential for rodent prevention were subject to a correlation analysis. For this purpose, the rank correlation coefficient according to Spearman was calculated. In order to compare the grouping variable potential for rodent prevention, measured by the item “*Condition of the farm to maximally prevent rodents from settling*”, with the determinants, two groups were formed using the median. Therefore, two groups were formed and compared: farms with a high potential for rodent prevention and farms with a low potential for rodent prevention. The Mann–Whitney U test was used to compare the two groups on different independent variables. Differences in the two groups are interpreted as influencing factors on rodent prevention, and respective *p*-values are shown.

## 4. Results

### 4.1. Characterisation of the Sample

Of the 24 participants, 22 were male and 2 were female (Table 2). The sample corresponded to the distribution in North Rhine-Westphalia in terms of gender. About two thirds of the participants were between 35 and 54 years old. The level of education of project farmers is shown in Table 2. Regarding the characteristics age and education, the sample was younger and had a higher level of education than the average in North Rhine-Westphalia.

The sample comprised 16 pig farmers, including 9 sow farmers and piglet rearers, and 10 cattle farmers, 6 of which were dairy farmers. Furthermore, there were five poultry farmers in the sample, and three of them were keeping poultry as a side activity. The comparison with the average in North Rhine-Westphalia showed that significantly more farms kept pigs and fewer kept cattle (Table 3). The sample corresponded to the distribution in North Rhine-Westphalia in the characteristic number of farms with poultry and small ruminants.

### 4.2. Implementation of the Pest Control Measures during the Duration of the Project 

Four different pest control companies were selected by the project farmers (Table 4). In agreement with the farmers, the companies carried out different concepts and approaches to pest control (e.g., frequency of bait box inspections, control inside or exclusively outside buildings). Farmers were free to change pest controllers throughout the project. One farmer stopped working with the pest controller in the first year of the project. After a one-year term, two farmers chose another pest controller who was listed.

Farmers participating in the project provided diverse assessments of rodent pressure on their farms prior to the project, with a large portion categorizing the pressure as either “rather high” or “rather low”, while fewer reported it as “very low” or “very high”. The predominant method for rodent control was the use of anticoagulant rodenticides, with snap traps and live traps less commonly being utilized. Preventive measures, such as secure feed storage, building sealing, and eliminating or sealing rodent refuges, were employed but with decreasing frequency.

### 4.3. Results of the Descriptive Variables

Attitudes towards rodent control were generally favourable, with farmers acknowledging its significance in maintaining biosecurity and preventing resistance to anticoagulants. However, uncertainties existed regarding perceived behavioural control, with farmers expressing uncertainty about their control efficacy and acknowledging difficulties associated with rodent management. Factors impeding effective control included structural issues, environmental factors, and peculiar rodent behaviour.

Farmers exhibited varying levels of adherence to rodent-control regulations. While there was a shared concern among farmers about disease transmission and structural damage caused by rodents, attitudes diverged regarding the risk to consumers. A significant proportion of farmers possessed certification permitting the use of anticoagulants, yet many expressed a lack of confidence in their training and desired additional guidance on best practices.

Moreover, farmers expressed a willingness to leverage project advisory services and embrace innovative methods, but they displayed limited contemplation about the advantages and disadvantages of participating in the project. Overall, the answers revealed a spectrum of attitudes and practices among farmers regarding rodent management, highlighting areas where further education and support could enhance rodent-control strategies on their farms.

#### 4.3.1. Core Variables of the Theory of Planned Behaviour (TpB)

Table 5 summarizes responses on core variables related to the TpB concerning rodent control. Farmers generally exhibited strong positive attitudes towards rodent control. They believe in rodent control protecting livestock from diseases and consider the importance of control measures. However, there were mixed sentiments regarding the perceived behaviour control, with high scores indicating a strong sense of responsibility for managing rodent pressure, yet lower scores suggesting uncertainty about complete control and acknowledgment of challenging conditions. Subjective norms were also positive, indicating high importance placed by family and employees on rodent control, while perceptions regarding monitoring, advisors’ influence, and neighbours’ interest varied. Overall, the mean values for attitudes, perceived behaviour control, and subjective norms highlighted were generally positive but slightly variable in inclinations among farmers regarding their attitudes, perceived control, and social influences concerning rodent control.

#### 4.3.2. Other Descriptive Variables on Likert-Scales

Table 6 presents farmers’ attitudes related to emotions, risk awareness, their need for information, and willingness to make changes or receive advice regarding rodent control. Farmers generally expressed discomfort and mild disgust towards rats and mice but exhibited a strong disinterest or dislike towards these rodents. In terms of risk awareness, farmers perceived higher risks associated with disease transmission to farm animals and structural damage caused by rodents, while perceiving comparatively lower risks regarding feed losses, harmful disease transmission to themselves and family, and transmission through animal products to consumers. Their expressed need for information indicated a desire for additional guidance on best practices in rodent control, although there were mixed feelings about feeling adequately informed. Farmers displayed a strong willingness to embrace advisory services and innovations associated with the project but showed some variability in contemplating the advantages and disadvantages of participating.

#### 4.3.3. Other Descriptive Variables

After one year, the project participants were asked about their satisfaction with the project. Overall, the likelihood of working with the pest controller after the end of the partial funding was at a high level (Table 7). Furthermore, project farmers were satisfied with the success of the pest control measures at this point in time of the project. Satisfaction with the success of the rodent-control measures was also positively related to the likelihood of further cooperation (r = 0.610, *p* < 0.01).

In the evaluation of objective knowledge (for questions see Table A1 in Appendix A), no participant answered all six questions correctly. Furthermore, 58% correctly answered three out of six questions and 42% of farmers answered four or five questions correctly. Especially, the two questions about the authorised persons in the use of anticoagulants caused difficulties for the farmers. There was also little knowledge about possible anticoagulant resistance in the region. We found that 83% of the farmers mentioned diseases/pathogens that can be transmitted by harmful rodents such as hantaviruses (44% of the farmers), followed by salmonella (26%), African swine fever (17%), and influenza (13%). The two questions on whether dogs and cats are adequate natural enemies to control mice and rats were answered correctly as being not true by most farmers (91% and 87%).

The question about the motives for participating in the project was asked as a ranking question. The most important reason for participating in the project was the partial financing (Table 8). Also, from the farmers’ point of view, trying out professional rodent control and being able to hand over more control responsibility to the pest controller was a reason to participate in the project. On the other hand, farmers see the promotion of biosecurity as less important.

The question “*Who has advised you on rodent control issues so far?*” was a ranking question. Almost 30% of the farmers stated that they did not have an advisor for rodent control before the start of the project. The sales advisor was mentioned as the most important contact person, followed by the farm veterinarian (Table 9).

With regard to project participation, it can be stated that an exchange on the topic of rodent control was encouraged at the beginning of the project because almost all farmers stated that they had talked to different people about the project, whereby the largest share of conversations took place with their own neighbours, who may also be farmers (n = 21), followed by friends (n = 12). A total of 14 participants gave multiple answers to this question, indicating two groups of people, in particular neighbours and friends.

### 4.4. Correlations between the Determinants 

The possible determinants that may be related to the potential for rodent prevention were tested for correlations among each other. The evaluation showed that the determinant risk awareness correlated strongly and positively with the attitude of the respondents (r = 0.59, *p* = 0.002). The higher the risk of damage on the farm was assessed, the more important rodent control was to the respondents, and vice versa. Risk awareness was also negatively correlated with perceived subjective behaviour control. The correlation was at a medium level (r = −0.41, *p* = 0.048). This means that the higher the level of control of rodent pressure on the farm, the lower the perceived risk of damage due to rodent pressure, and vice versa. Furthermore, there was a positive correlation between the need for information and willingness to make changes on the farm and to accept advice (r = 0.56, *p* = 0.004). The higher the respondents’ need for information, the higher their interest in developing themselves and their farm within the framework of the project. Moreover, there was a negative correlation between the subjectively assessed rodent pressure before the start of the project and the willingness to change (r = −0.47, *p* = 0.023). The objectively determined level of knowledge, measured by the six knowledge questions, also correlated with the willingness to change (r = 0.49, *p* = 0.015). This indicated that the higher the knowledge of the respondents, the higher the willingness to change within the framework of the project. The level of knowledge also correlated with risk awareness (r = 0.41, *p* = 0.050). Thus, the higher the actual knowledge about the hazards, the higher the risk of disease transmission was rated (Table A2 in Appendix A). Furthermore, the evaluation revealed that the motive “partial financing” correlated with the attitude of the respondents (r = 0.49, *p* = 0.014). The motive “opportunity to try out professional rodent control” also correlated with attitude (r = −0.43, *p* = 0.036). The motive “collaborative concept of rodent control” correlated negatively with the subjectively assessed rodent pressure before the start of the project (r = −0.50, *p* = 0.012). The lower the rodent pressure on their own farm was estimated to be, the more important the aspect of collaborative control was to the respondents.

### 4.5. Potential for Rodent Prevention

During the farm visits after 1.5 years of project duration, after a stable trust basis for on-site visits had been established, a person not involved in the project assessed rodent pressure as well as the potential for rodent prevention. The potential for prevention, measured by whether rodents were prevented from settling as much as possible, was described as good for more than 50% farms (median = 2) (Table 10).

The potential for rodent prevention was used as a categorical variable in the further analysis. Two groups were formed on the basis of the median. There were significant differences between the two groups only regarding the willingness to change (U = 29.500, *p* = 0.021). The participants with high potential for prevention were more willing to make changes at the beginning of the project (Table 11). There were no differences between the two groups regarding the motives for participating in the project. Farm-manager-related factors, such as age (U = 45.5, *p* = 0.225), education (χ(5) = 5.860, *p* = 0.387), and choice of pest controller (χ(3) = 0.619, *p* = 0.926) also had no effect. However, differences occurred depending on the type of animal kept. Thus, the potential for rodent prevention was found to be poor when cattle were kept on the farm (χ(1) = 5.064, *p* = 0.04).

## 5. Discussion

Our observational case study provides an assessment of possible attitudinal determinants to the potential for rodent prevention on livestock farms within an extended framework of the theory of planned behaviour (TpB). At the project’s onset, farmers had on average a self-assessed rodent infestation at a medium level [20]. Different measures advised by and implemented in conjunction with a professional pest controller were taken on all farms, ranging from the sole application of anticoagulant rodenticides to the use of combined measures. These included effective preventive measures, e.g., the securing of feed storage and buildings [10,57]. During the farm visits carried out after 1.5 years of the project, it became clear that the rodent pressure was low on the majority of the farms at this point in time. About half of the farms were very-well or well positioned with regard to the prevention of a (new) settlement of rodents. Some farms showed deficits.

Farmers take measures when the rodent pressure on their farms is above a subjective threshold. The willingness to change played a central role in our study. It was the only attitudinal determinant that was significantly higher in the group of farmers with higher rodent-prevention potential. Willingness to change was correlated both to the subjectively assessed rodent pressure at the beginning of the project and to objective knowledge. Farmers whose farms appeared tidy during the farm visits had signalled at the beginning of the project that they were open to innovations during their participation in the project and were happy to accept advice. At the beginning of the project, farmers were aware that rodents can cause health as well as infrastructural and economic damages. So, there was a fundamental interest from farmers in the successful implementation of rodent-control measures [25]. Although farmers stated that they can protect their livestock from epizootics through rodent control, the promotion of biosecurity only played a subordinate role in the motivation for participating in the project. This might be linked to a more pessimistic attitude towards the general term biosecurity related to health issues [35] versus a more pragmatic and concrete attitude on rodent control.

The most frequently mentioned motivation for participating in the project was partial financing. This raises questions on the economic sustainability of the chosen approach as co-financing to larger numbers of farms over longer periods of time does not seem viable under the current institutional framework. With high sectorial or societal benefits, it might be feasible to implement publicly (co-)financed rodent-control measures [58]. Financial costs caused by a professional pest controller played a central role to project farmers. The reduction in partial financing over time was seen as a matter of concern by some farmers in the project. Many studies on biosecurity have shown that farmers are more willing to implement measures on their farms if cost–benefit analyses show positive effects [35,40,43]. Valeeva et al. [59] found that penalties were more motivating than premiums. Farmers in our study seemed to expect too little benefit while costs were classified as high in hiring a pest controller. Further research is needed to explore the specific link between the perceived benefits of rodent control, the importance of partial financing, and the relevance of different cost categories, e.g., direct financial costs of pest controllers, opportunity costs of time as economic costs, and mental load as psychological costs.

The perceived social pressure by the professional environment to carry out rodent-control measures was moderate according to the theory of planned behaviour (TpB). Although rodent control is mandatory, few farmers reported regular inspections. Sighting of rats causes discomfort among respondents and an infestation was also an issue within the family and among employees [36]. At the start of the project, it was less important to neighbours whether rodents were controlled. Yet, the project stimulated exchange among neighbouring farms. Cooperative behaviour was increased and incentives for free riding were reduced when an individual felt part of a group [60].

A lack of professional pest controllers could be an additional problem [61]. A quarter of the farmers stated that there was no advice regarding rodent control before the project. Advisory services were provided by sales advisors and pest controllers as well as farm extension services. Yet, the focus of advisory services provided by sales advisors and farm consultants is essentially different. Veterinarians rarely advise farmers specifically on rodent-control issues [22]; although, veterinarians can play a central role in increasing implementation of biosecurity measures [36,37,62,63]. How to effectively include veterinarians should be further investigated.

With regard to perceived behaviour control within the theory of planned behaviour (TpB) framework, farmers considered themselves responsible [64]. Several obstacles were usually unavoidable like rodent-promoting geographical conditions from the immediate environment of the farms. Livestock farms with open stables can hardly be designed to be inaccessible to rodents [65,66,67]. Cattle farms performed comparatively worse. Preventive measures imply a considerable financial cost as structural measures like the renewal of doors and gates and the sealing of false ceilings and walls are costly. The fact that the farmers’ permanent attention is necessary to control rodent pressure can also be an inhibiting factor. Farmers mentioned the time saved by hiring a pest controller as motivation to participate in the project as well as to outsource rodent control. A desire to be able to transfer responsibility to an external service provider reflects a time-saving factor and can increase willingness to cooperate [20].

Farmers’ risk awareness with regard to negative consequences caused by a rodent infestation was high, in particular, regarding disease transmission to livestock [68,69]. This might have been related to the recruitment process for the project. The extent to which a person can reliably assess risks depends on the availability of practical information [70], as indicated by the the positive correlation between risk awareness and knowledge. Hantavirus infections were reported in the general press, e.g., [71]. In the project district, there were 5 cases in 2018 (235 cases nationwide) in which hantaviruses were detected in humans [72]. The most important pathogens, such as salmonella, clostridia, leptospires, and *E. coli* [4,73,74], were rarely mentioned by the farmers. Farmers were also aware of the risk of structural damage and feed losses due to rodents. Therefore, the risk awareness of farmers is a motivating factor and contributes positively to the implementation of rodent-control measures [75].

The most commonly used anticoagulant rodenticides on farms were second-generation anticoagulant rodenticides (SGARs), which belong to the so-called PBT substances (persistent, bioaccumulative, and toxic) [57]. Due to high environmental risks, they are subject to specific use restrictions, i.e., SGARs may only be used by professional users who are properly qualified. Qualified professional users also include farmers who have a certificate of competence according to the German Plant Protection Certificate Regulations. This applied to three quarters of the farmers surveyed. If the expert certificate is not available, farmers need a professional pest controller for control with SGARs. Many farmers stated that the use of anticoagulants in rodent control was not part of their formal education nor of their training for the certificate of competence in plant protection. The farmers did not feel well educated on the use of anticoagulants [76,77] and would have liked to have additional information material. Farmers might not have been aware of risks to non-target species either and should be trained to remove dead rats in routine searches during and after the application of anticoagulant rodenticides [78]. This asks for more extensive addressing of rodent prevention in the basic education of farmers but also in further training to sustain their qualification for pesticide applications. This might also be a reason for farmers to seek assistance from veterinarians or qualified advisors [79].

## 6. Limitations and Perspectives of the Study

Our study is not without limitations. In contrast to other studies using the theory of planned behaviour (TpB), it was not the behavioural intention but the result of the actual behaviour that was used as dependent variable [28]. In order to evaluate this behaviour objectively, an expert person independent of the project was deliberately consulted. The potential for rodent prevention was measured by the farms’ on-site conditions. However, it is not known in detail to what extent these farms had already implemented preventive measures at the beginning of the project, nor exactly what measures they had implemented during the first 1.5 years of the project. More detailed farmer-implemented rodent prevention measures should be surveyed in future studies.

Further limitations refer to our sample and the generalizability of the results. There are deviations in the sample from the distribution in North Rhine-Westphalia (NRW). As recruitment for the project raised awareness of rodent control, farmers probably had better prior knowledge than farmers in North Rhine-Westphalia (NRW) in general. Self-selection of farmers with high rodent pressure into the project might have further biased the sample. The size of project participation, with 24 out of 102 farms, appears to be small. Yet, the total of 102 farms also included those with low numbers of animals or animal husbandry as a hobby. The surveyed farms are the larger and more professional farms in the municipality. The comparability between the sample and a larger population is therefore limited. As this is an observational case study, results should be further validated in larger samples and more diverse locations.

## 7. Conclusions

The present observational case study confirms the importance of the implementation of preventive measures in the control of rodents on livestock farms. Thereby, willingness to change and knowledge play a central role. There is a need for action to provide farmers with more practical information on rodent control and prevention measures. Education, training, extension services, advice, and support for farmers is, therefore, essential to increasing the implementation of successful control and prevention measures within a one health approach. More practical information is also required to ensure that the more potent second-generation anticoagulant rodenticides (SGARs) are handled responsibly to reduce the impact on non-target species.

## Figures and Tables

**Figure 1 animals-13-03809-f001:**
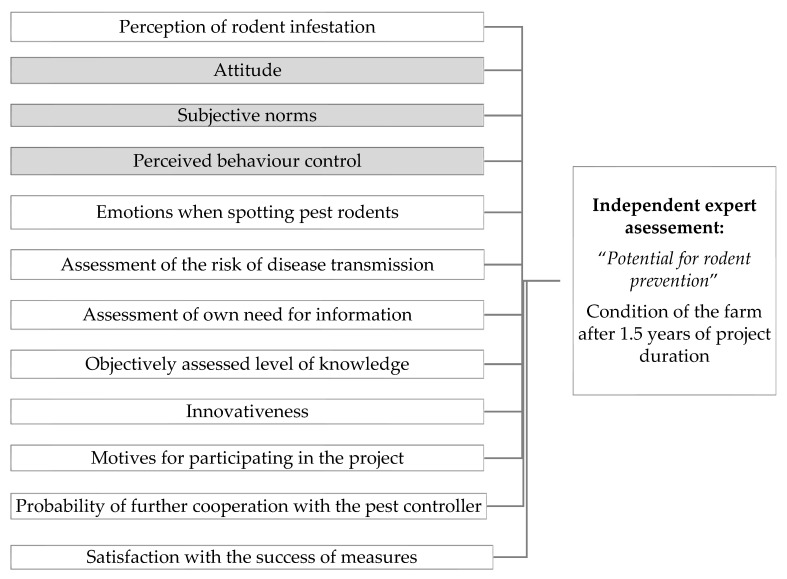
The underlying model used in this study to explain the potential for rodent prevention based on the theory of planned behaviour according to Ajzen [32] in grey and supplementary determinants in white assessed by farmers at project start and “*potential for rodent prevention*” assessed by independent expert after 1.5 years.

**Table 1 animals-13-03809-t001:** Minimum animal number per farm considered in the selection of the project municipality.

Farm Type	Minimum Animal Numbers
Pig farms	>50 animals
Cattle farm	>50 animals
Poultry farm	>350 animals
Broiler farms	>5000 animals
Sheep and goat farms	>100 animals

**Table 2 animals-13-03809-t002:** Comparison of the sample with the state-wide average in North Rhine-Westphalia in the characteristics gender, age and education.

Characteristic	Sample (n = 24)	North Rhine-Westphalia (n = 24,638)
**Gender ***		
male	91.7%	91.4%
female	8.3%	8.6%
**Age ***		
under 25 years	0.0%	0.0%
25–34 years	8.3%	6.2%
35–44 years	37.5%	17.1%
45–54 years	37.5%	36.6%
55–64 years	12.5%	31.9%
over 65 years	4.2%	7.7%
**Education ^#^**		
Vocational school, vocational training	4.2%	14.9%
Agricultural college	12.5%	11.5%
Further training, professional agricultural manager (*Fachagrarwirt*)	12.5%	10.1%
Secondary agricultural school, technical school	45.8%	15.0%
University, university of applied sciences	8.3%	5.0%
No agricultural vocational qualification	16.7%	43.5%

* Source NRW: Destatis 2017a [54]. ^#^ Source NRW: Destatis 2012 [55].

**Table 3 animals-13-03809-t003:** Comparison of the sample with the state-wide average in North Rhine-Westphalia with regard to the type of animal kept and production; absolute frequencies in brackets.

Animal Species	Sample (n = 24)	NRW (n = 24,638)
**Pigs**	**66.7% (16)**	**34.1%**
Sows and piglet rearing	37.5% (9)	-
Closed system	25.0% (6)	-
Pig fattening	8.3% (2)	-
**Cattle**	**41.7% (10)**	**56.8%**
Dairy cattle	25.0% (6)	-
Bull fattening	12.5% (3)	-
Suckler cows	4.1% (1)	-
**Poultry**	**20.8% (5)**	**19.6%**
Broilers	8.3% (2)	-
Poultry as a hobby	12.5% (3)	-
**Small ruminants**	**12.5% (3)**	**12.2%**
**Horses**	**12.5% (3)**	**22.9%**

Source NRW: Destatis 2017b [56].

**Table 4 animals-13-03809-t004:** Frequency distribution of pest controllers hired by farmers; absolute frequencies in brackets.

Pest Controllers	Sample at the Beginning of the Project ^a^	Sample after One Year ^b^	Sample after 1.5 Years ^c^
A	37.5% (9)	39.1% (9)	43.5% (10)
B	37.5% (9)	34.8% (8)	26.1% (6)
C	16.7% (4)	17.4% (4)	21.7% (5)
D	8.3% (2)	8.7% (2)	8.7% (2)
Total	100% (24)	100% (23)	100% (23)

^a^ March 2019 (24 project farmers were interviewed). ^b^ January 2020 (24 project farmers were interviewed). ^c^ August 2020 (farm visits took place at 23 project farms).

**Table 5 animals-13-03809-t005:** Results of the core variables of the theory of planned behaviour by statements to assess rodent management on farms in a municipality in Germany (1 = disagree at all to 10 = agree completely) (n = 24).

**Attitudes towards rodent control**	**Median**	**Mean ± SD**
By controlling rodents, I can protect my livestock from animal diseases.	10.0	9.04 ± 1.23
Avoiding resistance to anticoagulants is important for me.	9.5	8.66 ± 2.22
I give great importance to rodent control.	9.0	8.33 ± 1.88
Anticoagulants are dangerous to humans, non-target organisms and the environment.	8.0	7.50 ± 2.67
**Attitudes (mean value of all items)**		**8.39 ± 1.26**
**Perceived behaviour control**		
I am responsible for keeping the rodent pressure on my farm low.	10.0	9.21 ± 1.32
Rodent control is exclusively my responsibility.	7.0	7.17 ± 2.84
I have the rodent population on my farm under control.	6.0	6.42 ± 2.57
Rodent control on my farm is associated with difficult conditions. *	5.0	5.83 ± 3.10
**Perceived behaviour control (mean value of all items)**		**6.99 ±1.22**
**Subjective norms**		
It is important to my family that I control rodents.	10.0	9.25 ± 0.94
I am obliged to control rodents on my farm.	10.0	8.04 ± 2.99
It is important to my employees that I control rodents.	9.5	8.40 ± 2.32
I am regularly controlled whether I am taking measures against rodents.	7.5	7.00 ± 3.08
My advisors tell me that I need to control rodents.	5.5	5.79 ± 3.31
It is important to my neighbours that I control rodents.	3.0	4.08 ± 3.21
**Subjective norms (mean value of all items)**		**6.92 ± 1.40**

* These statements were reverse-scaled to calculate the mean value.

**Table 6 animals-13-03809-t006:** Results of the other variables on Likert-scales by statements to assess rodent management on farms in a municipality in Germany (1 = disagree at all to 10 = agree completely) (n = 24).

**Emotions**	**Median**	**Mean ± SD**
Rats and mice create discomfort in me.	10.0	8.17 ± 3.07
I am disgusted by rats and mice.	4.5	5.46 ± 3.68
I do not care when I see rats and mice. *	1.0	1.42 ± 1.44
I like rats and mice. *	1.0	1.17 ± 0.82
**Emotions (mean value of all items)**		**8.26 ± 1.62**
**Risk awareness: How high do you estimate the risk…**		
of disease transmission from rodents to your farm animals?	8.5	8.08 ± 2.21
of structural damage caused by rodents?	8.0	7.92 ± 1.64
of feed losses due to rodents?	7.5	6.50 ± 3.22
of the transmission of harmful rodents to you, your employees and your family?	7.5	5.96 ± 3.06
of disease transmission from harmful rodents through animal products to the end consumer?	3.0	4.33 ± 2.71
**Risk awareness (mean value of all items)**		**6.56 ± 1.83**
**Assessment of own need for information**		
I would like to have additional information on “best practice” in rodent control.	8.0	6.50 ± 3.11
I feel well informed about the “best practice” in rodent control. *	4.0	4.71 ± 2.77
I am well trained in the use of anticoagulants due to the certificate of competence in plant protection. *	3.0	3.61 ± 2.85
**Need for information (mean value of all items)**		**6.73 ± 2.19**
**Willingness in the course of the project to make changes or to receive advice**		
I am happy to benefit from the advisory services offered.	10.0	9.79 ± 0.83
I am open-minded to innovations associated with participation.	10.0	9.38 ± 1.06
I have thought carefully about the advantages and disadvantages.	8.5	8.29 ± 1.97
**Willingness to change (mean value of all items)**		**9.15 ± 1.01**

* These statements were reverse-scaled to calculate the mean value.

**Table 7 animals-13-03809-t007:** Probability of cooperation with the pest controller after the end of the project and satisfaction with the control success (1 = low agreement to 5 = high agreement) (n = 24).

	Median	Mean ± SD
After the project has ended, how likely are you to continue working with your pest controller?	4.0	3.35 ± 1.23
Overall, how satisfied are you with the success of the rodent control measures?	4.0	3.63 ± 1.17

**Table 8 animals-13-03809-t008:** Ranking question about motivation for participating in the project (rank 0 = not chosen to rank 8 = most important reason) (n = 24).

Motives for Participating	Median	Mean ± SD
Partial financing	7.0	6.00 ± 2.28
Collaborative concept of rodent control	6.0	5.29 ± 2.61
Opportunity to hand over rodent control completely to a pest controller	5.5	4.13 ± 3.69
Opportunity to try out professional rodent control	5.0	4.13 ± 3.38
Promoting biosecurity	3.0	2.87 ± 3.04
Knowledge gain	0.0	2.29 ± 2.91
Participation in a scientific project	0.0	1.13 ± 2.13
Other	0.0	0.13 ± 0.61

**Table 9 animals-13-03809-t009:** Ranking question on the advisory situation on the farms (n = 24) (rank 0 = not selected to rank 7 = most important advisor).

Advisory Services: Who Has Advised You on Rodent-Control Issues so Far?	Mean ± SD
Sales advisor	2.8 ± 3.4
Nobody	2.0 ± 3.3
Farm veterinarian	1.5 ± 2.8
Pest controller	1.2 ± 2.7
Agricultural consultant	0.9 ± 2.4
Other	0.7 ± 2.0
Consultant from the chamber of agriculture	0.2 ± 1.0

**Table 10 animals-13-03809-t010:** Potential for rodent prevention and rat pressure after 1.5 years of project duration, assessed by an independent third party (potential of rodent prevention: 1 = strongly agree and 5 = strongly disagree, rodent pressure: 1 = not present to 5 = very high) (n = 23).

Potential of Rodent Prevention and Rodent Pressure	Median	Mean ± SD
The farm is in such a condition that prevents rodents from settling to the maximum possible extent.	2.0	2.83 ± 1.07
How great is the rat pressure in the farm at the moment?	1.0	1.57 ± 0.79

**Table 11 animals-13-03809-t011:** Comparison of groups with low and high potential for rodent prevention as a function of potentially influencing determinants.

Dependent Variables	Low Potential for Rodent Prevention (n = 10)		High Potential for Rodent Prevention (n = 13)		*p*-Value
	Median	Mean ± SD	Median	Mean ± SD	
Attitude	8.4	8.5 ± 0.6	8.3	8.2 ± 1.6	0.901
Subjective norms	7.0	7.1 ± 1.2	7.2	6.7 ± 1.6	0.709
Behaviour control	6.8	6.7 ± 1.5	7.3	7.3 ± 0.9	0.248
Emotions	9.5	8.7 ± 1.6	8.0	7.8 ± 1.6	0.178
Risk awareness	6.5	6.3 ± 2.0	7.0	6.6 ± 1.8	0.756
Need for information	5.8	6.4 ± 2.6	7.0	7.0 ± 1.9	0.709
Willingness to change	8.8	8.6 ± 1.1	10.0	9.5 ± 0.7	0.021
Subjectively assessed rodent pressure	3.8	3.8 ± 0.6	3.0	3.3 ± 0.6	0.179
Objectively determined knowledge	3.0	3.1 ± 1.0	3.0	3.3 ± 1.0	0.474
Satisfaction with control	3.5	3.4 ± 1.4	4.0	4.0 ± 1.4	0.354
Probability of further cooperation with the pest controller	2.5	5.1 ± 4.0	3.0	3.5 ± 2.1	0.181

## Data Availability

Authors agree to make their data available upon reasonable request. It is up to the author to determine whether a request is reasonable.

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
