# Peer review of "Farmers’ Attitudes in Connection with the Potential for Rodent Prevention in Livestock Farming in a Municipality in North Rhine-Westphalia, Germany"

_animals, 2023, doi:10.3390/ani13243809_

Round 1
Reviewer 1 Report
Comments and Suggestions for Authors
This paper sets out a survey of intensive livestock farmers in Germany about their attitudes about rodents and about the benefits of enlisting support from pest control operators. The theory of planned behaviour model was used to assess attitudes and behaviours. This sets out a way to consider some important issues around rodent management in intensive livestock industries. This type of work is critical to develop effective strategies to manage rodents in these systems, and could provide a framework for others in similar situations in different countries.
Although the paper is well intentioned, unfortunately, I have several concerns about it. I provide some general comments below and make lots of comments and suggestions as comments directly on the PDF document.
1. The paper is missing a review of impacts of rodents to (intensive) livestock systems - why would undertaking this study will help?
2. What other assessments have been made about attitudes to rodent management? What insights do they provide and how can this help with designing better management? There is a body of work assessing the knowledge, attitudes and practices of farmers relating to rodent management now in many countries. I think it is an oversight to not include some of this as background information or in terms of frameworks to use.
3. The paper is very long.
a. I suggest that the Introduction be split, and provide a “Background Theory” section to explain the TPB. Suggest to compare TPB with other frameworks. Why choose TPB over other approaches?
b. The Results were too long and too detailed (very descriptive). There are lots of Tables (16 in all, plus 2 in the Appendix). Most of the text is a repeat of what is in the Tables. It is normally suggested to ensure the paper can be read without having to refer to the Tables, but I think in this case, many of the Tables could be combined, and then a higher-level synthesis of the key findings should then be provided. The details can be found in the Tables.
c. The Discussion was also too long. The Discussion lacked a connection back to the literature. The results from this analysis needs to be compared with findings from the literature. What was the same or different? What new things were learnt? What gaps remain? Was the framework a useful approach? What needs to be done to improve rodent management in these intensive systems?
4. It is not clear what services the pest controller actually provided to the livestock farmers. I think this is critical information. This can help define a minimum set of information that is required to improve management outcomes. How were they funded? Furthermore, there was no description of what the farmers have actually undertaken (over and above what they would have done anyway)?
5. More information is needed about the independent assessment of the livestock farms. What aspects were considered? An overall score? Or was there multiple aspects scored separately? I would be interested in the types of things assessed and how they were scored. More information about this would be useful.

Comments on the Quality of English LanguageI have made several suggestions in my comments on the PDF. The paper can be improved.
Author Response
Comments and Suggestions for Authors from Reviewer 1 and responses from authors
This paper sets out a survey of intensive livestock farmers in Germany about their attitudes about rodents and about the benefits of enlisting support from pest control operators. The theory of planned behaviour model was used to assess attitudes and behaviours. This sets out a way to consider some important issues around rodent management in intensive livestock industries. This type of work is critical to develop effective strategies to manage rodents in these systems, and could provide a framework for others in similar situations in different countries.
à Thank you for acknowledging the value of the manuscript.
Although the paper is well intentioned, unfortunately, I have several concerns about it. I provide some general comments below and make lots of comments and suggestions as comments directly on the PDF document.
à Thanks for the many and very detailed and very helpful comments. We have gladly taken them into account. All comments have been responded to in detail in the attached pdf.
- The paper is missing a review of impacts of rodents to (intensive) livestock systems - why would undertaking this study will help?
à impacts of rodents have been extended in the introduction
- What other assessments have been made about attitudes to rodent management? What insights do they provide and how can this help with designing better management? There is a body of work assessing the knowledge, attitudes and practices of farmers relating to rodent management now in many countries. I think it is an oversight to not include some of this as background information or in terms of frameworks to use.
à more studies are cited in the introduction and taken into account. Yet in order to make the manuscript not much longer, they were only included as short mentioning.
- The paper is very long.
à the results section has been shortened considerably. The discussion has been restructured.
- I suggest that the Introduction be split, and provide a “Background Theory” section to explain the TPB. Suggest to compare TPB with other frameworks. Why choose TPB over other approaches?
à Thanks for this valuable suggestion. We split the introduction and moved parts of it to a new section “Theoretical background”.
- The Results were too long and too detailed (very descriptive). There are lots of Tables (16 in all, plus 2 in the Appendix). Most of the text is a repeat of what is in the Tables. It is normally suggested to ensure the paper can be read without having to refer to the Tables, but I think in this case, many of the Tables could be combined, and then a higher-level synthesis of the key findings should then be provided. The details can be found in the Tables.
à results section has been shortened considerably. Accompanying text has been re-written completely. Tables have been merged.
- The Discussion was also too long. The Discussion lacked a connection back to the literature. The results from this analysis needs to be compared with findings from the literature. What was the same or different? What new things were learnt? What gaps remain? Was the framework a useful approach? What needs to be done to improve rodent management in these intensive systems?
à The discussion has been restructured by moving certain parts in an adapted manner to the section “limitations”. Additional references have been consulted and cited.
- It is not clear what services the pest controller actually provided to the livestock farmers. I think this is critical information. This can help define a minimum set of information that is required to improve management outcomes. How were they funded? Furthermore, there was no description of what the farmers have actually undertaken (over and above what they would have done anyway)?
à Missing information has been added in the revised version of the manuscript.
- More information is needed about the independent assessment of the livestock farms. What aspects were considered? An overall score? Or was there multiple aspects scored separately? I would be interested in the types of things assessed and how they were scored. More information about this would be useful.
à Missing information has been added in the revised version of the manuscript.
I have made several suggestions in my comments on the PDF. The paper can be improved.
à Thanks for the many and very detailed and very helpful comments. We have gladly taken them into account. All comments have been responded to in detail the attached pdf.

Reviewer 2 Report
Comments and Suggestions for Authors
General comments
I appreciate the authors for allowing me to review the proposal of their manuscript that addresses the issue of rodent control on farms and the attitudes of the producers themselves towards the control of these problems, which allows them to give a clear perspective on their limitations and areas for improvement of current strategies. Therefore, this article is innovative and suitable for publication in this journal. However, before its publication, it is necessary to make adjustments to improve its proposal.
Response:
Particular comments
Line 12. I suggest that they mention the range of years they considered for the study, if the authors consider it appropriate I suggest that this data be mentioned at the end of their objective in line 13.
Response.
Line 14. The authors could mention the direction of the correlation and the obtained value of the obtained correlation.
Response:
Line 15. If the authors allow me I suggest replacing the beginning of this statement from "Nearly half of the farms" to "X% of the farms showed good ....".
Response:
Line 19. I agree with your initial statement in your summary, however, if the authors will allow me I suggest that this statement be changed to "Rodents in production units are a threat to the health of production animals".
Response:
Line 22. In line with my comment made in line 12, please indicate the range of years considered in the analysis of the farms.
Response:
Lines 26 - 29. I understand the concern of the authors to comply with the number of words in their abstract, however, the description of their results are general, therefore I suggest that they mention the exact description of the results, for example, if the authors consider it appropriate, I would recommend that the statement on line 26 mention that "the farm visits show that rodent prevention potential presented a positive correlation (r=xx) with rodent pressure". Similarly, I would recommend specifying the significant differences on participants with a high and a low potential for rodent prevention.
Response:
Line 36. I recommend that authors can include "One Health" within keywords that may increase the chances of database hits.
Response:
Line 40. Please, add reference.
Response:
Line 41. Please, I suggest correcting the description of the scientific name of "Rattus norwegicus" to "Rattus norvegicus", likewise in this same statement you could just mention some examples of infectious diseases that can be transmitted to farm animals. I understand your concern for humans, however, I suggest that a narrower focus on the impact on production units could be given.
Response:
Line 42. I invite authors to review the authors' guide on the citation format recommended by the journal, since according to their patched citation format it would be [1-3, 4].
Response:
Line 58. I suggest integrating this paragraph with the next one as it gives continuity to your idea.
Response:
Line 60. In addition to my comment made on line 42, I suggest that you review the citation format of the references in the text.
Response:
Lines 115 - 124. This paragraph is confusing and does not help to understand the real relationship on the broad basis discussed in previous paragraphs. If the authors will allow me to suggest that the idea of lines 120 - 124 be replaced by a hypothesis where they point out that possible interaction of producer attitudes may influence rodent prevention.
Response:
Lines 126 - 128. The idea shown in this paragraph does not describe in a general way the data of the characteristic of your study, I suggest that you could include in this paragraph if your study is a retrospective or prospective study, randomized or blinded, also in this section it would be necessary that you mention the years where you collected the data of your study.
Response:
Line 148. As mentioned with the citation format, I recommend that authors review the citation format for tables or figures in the authors' guide, since they are regularly cited as (Table 1).
Response:
Line 208. Please mention if the significance level considered was p<0.05.
Response:
Lines 444 - 449- The initial paragraph I consider to be weak and the discussion made for the subsequent paragraphs is inconsistent, I suggest that you describe in a general way the most relevant results in your study, for example, if it is possible to mention the overall mean of the attitudes towards rodent control or the perceived sujective norms.
Response:
Lines 450 - 456. This paragraph describes the limitations of your study, if my interpretation is correct I suggest that this paragraph be relocated to the end of this section in a more comprehensive manner.
Response:
Line 577. Please specify which example you are referring to.
Response:
Line 580 Please specify the name and species of the specific pathogen, also cite it in italics.
Response:
Line 597. This statement is very interesting and I suggest that it could be expanded because I consider that educational constraints is an important constraint, however, here I suggest mentioning the general educational level of the producers, in addition it could be expanded where it is discussed that this is a reason for the farmer to seek assistance from veterinarians or qualified advisors, but the availability of experts makes management difficult. If you agree with my idea please include it.
Response:
Author Response
Comments and Suggestions for Authors from Reviewer 2 and responses from authors
General comments
I appreciate the authors for allowing me to review the proposal of their manuscript that addresses the issue of rodent control on farms and the attitudes of the producers themselves towards the control of these problems, which allows them to give a clear perspective on their limitations and areas for improvement of current strategies. Therefore, this article is innovative and suitable for publication in this journal. However, before its publication, it is necessary to make adjustments to improve its proposal.
à Thanks for the many and very detailed and very helpful comments. We have gladly taken them into account. All comments have been responded to below. Revision can be followed up by the track-change-function.
Particular comments
Line 12. I suggest that they mention the range of years they considered for the study, if the authors consider it appropriate I suggest that this data be mentioned at the end of their objective in line 13.
à reference has been made to the time frames and years in the simple summary. More details are provided in the methods section of the full paper.
Line 14. The authors could mention the direction of the correlation and the obtained value of the obtained correlation.
à Authors think that this would be too much specific information in the simple summary. Exact figures are given in the results section.
Line 15. If the authors allow me I suggest replacing the beginning of this statement from "Nearly half of the farms" to "X% of the farms showed good ....".
à Due to the small simple size, giving percentages in the simple summary seems inappropriate to the authors. Details are given in the results section.
Line 19. I agree with your initial statement in your summary, however, if the authors will allow me I suggest that this statement be changed to "Rodents in production units are a threat to the health of production animals".
à Authors prefer the term “livestock” and the reference to “one health” in order emphasize the wider relevance of the topic which reaches beyond animal production.
Line 22. In line with my comment made in line 12, please indicate the range of years considered in the analysis of the farms.
à reference made to the year.
Lines 26 - 29. I understand the concern of the authors to comply with the number of words in their abstract, however, the description of their results are general, therefore I suggest that they mention the exact description of the results, for example, if the authors consider it appropriate, I would recommend that the statement on line 26 mention that "the farm visits show that rodent prevention potential presented a positive correlation (r=xx) with rodent pressure". Similarly, I would recommend specifying the significant differences on participants with a high and a low potential for rodent prevention.
à Authors prefer not to give numbers and only summarize results as selection of a very limited number of numerical results would be arbitrarily.
Line 36. I recommend that authors can include "One Health" within keywords that may increase the chances of database hits.
à “one health” included in keywords.
Line 40. Please, add reference.
à reference is made to review articles at the end of the paragraph
Line 41. Please, I suggest correcting the description of the scientific name of "Rattus norwegicus" to "Rattus norvegicus",
à corrected
likewise in this same statement you could just mention some examples of infectious diseases that can be transmitted to farm animals. I understand your concern for humans, however, I suggest that a narrower focus on the impact on production units could be given.
à examples of transmitted diseases have been added. In order to show a broader relevance of rodent control, authors prefer to also mention transmission to pets and humans.
Line 42. I invite authors to review the authors' guide on the citation format recommended by the journal, since according to their patched citation format it would be [1-3, 4].
à will be corrected in the final editing
Line 58. I suggest integrating this paragraph with the next one as it gives continuity to your idea.
à done
Line 60. In addition to my comment made on line 42, I suggest that you review the citation format of the references in the text.
à will be corrected in the final editing
Lines 115 - 124. This paragraph is confusing and does not help to understand the real relationship on the broad basis discussed in previous paragraphs. If the authors will allow me to suggest that the idea of lines 120 - 124 be replaced by a hypothesis where they point out that possible interaction of producer attitudes may influence rodent prevention.
à the whole paragraph has been revised based on comments from both reviewers
Lines 126 - 128. The idea shown in this paragraph does not describe in a general way the data of the characteristic of your study, I suggest that you could include in this paragraph if your study is a retrospective or prospective study, randomized or blinded, also in this section it would be necessary that you mention the years where you collected the data of your study.
à Based on comments of reviewer 1, this section has been re-arranged completely. Thereby we also included this comment from reviewer 2.
Line 148. As mentioned with the citation format, I recommend that authors review the citation format for tables or figures in the authors' guide, since they are regularly cited as (Table 1).
à will be corrected in the final editing
Line 208. Please mention if the significance level considered was p<0.05.
à instead of using a specific significance level, authors decided to display p-values so readers can make their own choice regarding the level of significance.
Lines 444 - 449- The initial paragraph I consider to be weak and the discussion made for the subsequent paragraphs is inconsistent, I suggest that you describe in a general way the most relevant results in your study, for example, if it is possible to mention the overall mean of the attitudes towards rodent control or the perceived subjective norms.
à the paragraph and the subsequent discussion have been revised
Lines 450 - 456. This paragraph describes the limitations of your study, if my interpretation is correct I suggest that this paragraph be relocated to the end of this section in a more comprehensive manner.
à This has been shifted to a new section “6. Limitations”
Line 577. Please specify which example you are referring to.
à the referenced source can be consulted for more details. Authors do not deem necessary to provide more specific details here as zoonotic transmission to humans is not the main focus of the paper.
Line 580 Please specify the name and species of the specific pathogen, also cite it in italics.
à as we refer to mentioning of farmers in the survey and as they did not give more details, authors do not deem necessary to give details that go beyond this information.
Line 597. This statement is very interesting and I suggest that it could be expanded because I consider that educational constraints is an important constraint, however, here I suggest mentioning the general educational level of the producers, in addition it could be expanded where it is discussed that this is a reason for the farmer to seek assistance from veterinarians or qualified advisors, but the availability of experts makes management difficult. If you agree with my idea please include it.
à Thanks for this helpful comment which we gratefully included in the discussion.
Round 2
Reviewer 1 Report
Comments and Suggestions for Authors
The authors have done a good job of addressing my concerns from the first version. I have some general comments and a few minor suggestions (provided on the PDF version of the paper).
General comments:
1. The paper is still far too long in my opinion (27 pages). I’m not sure what the suggested length should be for these papers in Animals, but I would thought 15-20 pages would be sufficient. The section on Limitations could be shortened.
2. I make some comments on the PDF about what services the pest controllers are providing. There is information buried in the text of the paper, but it would be useful to articulate what was provided earlier in the paper to help provide better context.
3. Please check the tense throughout. It should be in past tense. I highlighted some instances, but there are probably many more.

Comments on the Quality of English LanguageSee comments above and on the PDF
Author Response
Comments and Suggestions for Authors from Reviewer 1 and responses from authors
The authors have done a good job of addressing my concerns from the first version. I have some general comments and a few minor suggestions (provided on the PDF version of the paper).
General comments:
- The paper is still far too long in my opinion (27 pages). I’m not sure what the suggested length should be for these papers in Animals, but I would thought 15-20 pages would be sufficient. The section on Limitations could be shortened.
--> we tried hard to reduce the length and checked each word and sentence if they are really necessary. In several cases, this was possible throughout the whole article.
- I make some comments on the PDF about what services the pest controllers are providing. There is information buried in the text of the paper, but it would be useful to articulate what was provided earlier in the paper to help provide better context.
--> all comments from the pdf have been included.
- Please check the tense throughout. It should be in past tense. I highlighted some instances, but there are probably many more.
--> done

Reviewer 2 Report
Comments and Suggestions for Authors
General comments
I appreciate the authors for having considered my previous comments on their manuscript, which I consider has an important potential in the area and provides very innovative information. However, I still feel that minimal adjustments need to be made to its presentation.
Response:
Particular comments
Simple summary and abstract. I understand the concerns of the authors to comply with the requirements of the authors' guidelines for publication, however, I still consider that for a reader who is interested in your article, it would be necessary to have basic data to help increase the impact of your results. If the authors agree, I suggest that the information requested in my previous comments should be integrated, in support of which I add the below:
Line 12. I suggest that they mention the range of years they considered for the study, if the authors consider it appropriate I suggest that this data be mentioned at the end of their objective in line 13.
Response:
Line 14. The authors could mention the direction of the correlation and the obtained value of the obtained correlation.
Response:
Line 15. If the authors allow me I suggest replacing the beginning of this statement from "Nearly half of the farms" to "X% of the farms showed good ....".
Response:
Line 616. I would appreciate it if you could add a section on the limitations of your study, however, I would recommend that this section be mentioned as "limitations and perspectives of study" if my interpretation is correct, many of these limitations are areas for future study. Otherwise if you agree I suggest adding a short paragraph stating these future trends.
Response:
Author Response
Comments and Suggestions for Authors from Reviewer 2 and responses from authors
General comments
I appreciate the authors for having considered my previous comments on their manuscript, which I consider has an important potential in the area and provides very innovative information. However, I still feel that minimal adjustments need to be made to its presentation.
--> We are grateful for the further comments and are happy to include them.
Particular comments
Simple summary and abstract. I understand the concerns of the authors to comply with the requirements of the authors' guidelines for publication, however, I still consider that for a reader who is interested in your article, it would be necessary to have basic data to help increase the impact of your results. If the authors agree, I suggest that the information requested in my previous comments should be integrated, in support of which I add the below:
Line 12. I suggest that they mention the range of years they considered for the study, if the authors consider it appropriate I suggest that this data be mentioned at the end of their objective in line 13.
--> We have included this information here and deleted it in the next sentence.
Line 14. The authors could mention the direction of the correlation and the obtained value of the obtained correlation.
--> We decided not to refer to correlations in the summary and abstract any more.
Line 15. If the authors allow me I suggest replacing the beginning of this statement from "Nearly half of the farms" to "X% of the farms showed good ....".
--> As explained before, authors do not deem it advisable to put percentages in the abstract of a study with a case study character with a limited sample size.
Line 616. I would appreciate it if you could add a section on the limitations of your study, however, I would recommend that this section be mentioned as "limitations and perspectives of study" if my interpretation is correct, many of these limitations are areas for future study. Otherwise if you agree I suggest adding a short paragraph stating these future trends.
--> We happily take this recommendation and we have changed the section’s heading